# Time dependent variational principle for tree tensor networks

**Daniel Bauernfeind⋆ and Markus Aichhorn**

Institute of Theoretical and Computational Physics,
Graz University of Technology, 8010 Graz, Austria

⋆ daniel.bauernfeind@tugraz.at

## Abstract

We present a generalization of the Time Dependent Variational Principle (TDVP) to any finite sized loop-free tensor network. The major advantage of TDVP is that it can be employed as long as a representation of the Hamiltonian in the same tensor network structure that encodes the state is available. Often, such a representation can be found also for long-range terms in the Hamiltonian. As an application we use TDVP for the Fork Tensor Product States tensor network for multi-orbital Anderson impurity models. We demonstrate that TDVP allows to account for off-diagonal hybridizations in the bath which are relevant when spin-orbit coupling effects are important, or when distortions of the crystal lattice are present.


# 1  Introduction

The development of the Density Matrix Renormalization Group (DMRG) [1, 2] was an immensely important milestone in our understanding of one-dimensional quantum systems. The subsequent realizations that DMRG produces Matrix Product States [3] (MPS) and that it can be formulated as a variational method [4], ultimately led to the development of numerous approaches using not only MPS but also general Tensor Networks to handle quantum systems. Notable examples are the Projected Entangled Pair States (PEPS) [5, 6], the Multi-scale Entanglement Renormalization Ansatz (MERA) [7] and so-called Tree-Tensor Networks (TTN) [8–17] including also the recently developed Fork Tensor Product States (FTPS) method [18, 19].

Among the most important properties of tensor networks is whether their graph is loop-free, i.e., whether there exists only a single path from one tensor to any other. While PEPS and MERA are not loop-free, the TTNs and MPS are. Cutting any edge of a loop-free network, results in two separated segments and therefore gives a notion of left and right with respect to this edge. This in turn allows a controlled truncation scheme based on the Schmidt-decomposition of quantum states in the spirit of DMRG.

One of the major reasons behind the success of tensor networks are the celebrated area laws of entanglement [20] stating that the entanglement of ground states of gapped Hamiltonians with short-range couplings is proportional to the surface area connecting the two regions. MPS in 1-d and PEPS in 2-d efficiently encode quantum states obeying these area laws and are hence efficient parametrizations. In addition, MPS-based time evolution for one dimensional systems is an important method to calculate dynamical properties [21–24]. Approaches to perform the real-time evolution include, among others, the time-dependent Density Matrix Renormalization Group (tDMRG) [25, 26], the closely related Time Evolving Block Decimation (TEBD) [27, 28] as well as the Time Dependent Variational Principle (TDVP) [29–31]. An in-depth comparison of several time evolution algorithms performed in Ref. [32] came to the conclusion that while all approaches have strengths and weaknesses, TDVP is among the most reliable methods to perform the time evolution.

While time evolution approaches for MPS are well established, much less has been done for general tensor networks. So far, mostly TEBD (and variations) have been used, for example for the MERA network [33], for PEPS [34–36] and for TTNs [9, 17, 18, 37, 38]. The advantage of TEBD is its relative simplicity, since it effectively boils down to a repeated application of short range operators obtained from a Suzuki-Trotter decomposition [39] of the full time-evolution operator.

However, one of the major disadvantages of TEBD is that it can become difficult to implement for more complicated Hamiltonians, especially when long-range couplings are present. One approach to treat such couplings is an MPO-based approach introduced by Zaletel et al. [44] in which an MPO approximation of the time-evolution operator is constructed. Alternatively, TDVP circumvents this problem by only demanding a Hamiltonian represented in the same tensor network structure as the state which is often easy to find. Additionally, TDVP in its single-site variant exactly respects conserved quantities of the Hamiltonian like energy or magnetization [30]. Although some works applied TDVP to more general tensor networks [45, 46], it is not obvious how these algorithms work in detail and how it can be generalized. A notable exception is Ref. [47] which introduces TDVP for binary TTNs. Parallel to these developments in the tensor network physics community, very similar approaches to TDVP have been developed in quantum chemistry under the name of Multi-layer Multi-Configurational Time-Dependent Hartree approach [40–43]. These methods effectively generate tensor networks by repeatedly grouping degrees of freedom together and transforming them with (time dependent) basis transformations into new degrees of freedom.

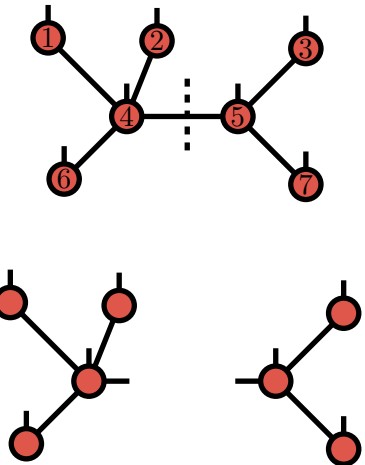

Figure 1: Example of a TTN with 7 tensors with different numbers of link-indices on each site. Each dot represents a tensor and each line an index, where a connected line implies summation over this index. The open lines are the physical indices $s_1 \cdots s_N$ ($N = 7$), while the connected lines are the link indices $q_1 \cdots q_L$ ($L = 6$). Cutting the link between sites 4 and 5, as indicated by the dashed line, results in two disconnected tensor network segments and defines a notion of left and right at each link. In this example, sites 1, 2, 3, 6 and 7 are the leaves of the TTN.

A more practical motivation for the formulation of TDVP for TTNs are Dynamical Mean-Field Theory (DMFT) calculations using the FTPS tensor network. So far, this approach has been used for so-called diagonal hybridizations only. On the other hand, real materials often exhibit off-diagonal hybridizations, which can for example come from spin-orbit coupling, or from distortions of the crystal lattice. For off-diagonal hybridizations, the TEBD approach used so far [18, 19] is difficult to generalize and we hence choose to use TDVP in these situations.

Although part of the motivation for this work comes from the FTPS tensor network, in this paper we formulate TDVP for *general* loop-free and finite-size tensor networks. After establishing the relevant concepts of TTNs in Sec. 2, we generalize TDVP to these networks in Sec. 3. Finally in Sec. 4 we show how this approach can be used for the FTPS tensor network and that it can be applied to off-diagonal hybridizations.

## 2 Tree Tensor Networks Basics

In this section, we discuss concepts of TTNs relevant for the formulation of TDVP. All these properties are generalizations of the corresponding concepts for MPS. Although these have been discussed previously in several publications (see for example Refs. [14, 17, 48]), here, we present them in a format that will suit us for the subsequent formulation of the TDVP algorithm.

### 2.1 TTNs

Any state $|\psi\rangle$ of a quantum system consisting of $N$ sites with local basis states $|s_i\rangle$ on site $i$ can be expanded in the corresponding product basis:

$$|\psi\rangle = \sum_{s_1 \cdots s_N} c_{s_1 \cdots s_N} |s_1 \cdots s_N\rangle. \tag{1}$$

Figure 2: Gauge degree of freedom in tensor networks. At each link, one can insert an identity $\mathbb{1} = G \cdot G^{-1}$ without changing the physical state $|\psi\rangle$. By absorbing $G$ into one tensor and $G^{-1}$ into the other, we obtain a different representation of the same state $|\psi\rangle$.

The coefficient $c_{s_1 \cdots s_N}$ is interpreted as a rank-N tensor with indices $s_1 \cdots s_N$. Tensor networks represent this rank-N tensor as a product over tensors of much smaller rank:

$$c_{s_1 \cdots s_N} = \sum_{q_1 \cdots q_L} T_{Q_1}^{s_1} \cdot T_{Q_2}^{s_2} \cdots T_{Q_N}^{s_N}$$

$$|\psi[T]\rangle = \sum_{\substack{s_1 \cdots s_N \\ q_1 \cdots q_L}} T_{Q_1}^{s_1} \cdot T_{Q_2}^{s_2} \cdots T_{Q_N}^{s_N} |s_1 \cdots s_N\rangle. \tag{2}$$

Each tensor $T_{Q_i}^{s_i} \equiv T_{q_1 q_2 \cdots q_{r_i}}^{s_i}$ has a set of auxiliary indices $Q_i = \{q_k : q_k \text{ is attached to node } i\}$ such that each auxiliary index is part of exactly two tensors. We call $r_i = |Q_i|$ the number of indices of the tensor on site $i$. Additionally, we attached to each tensor a physical index as for example in the FTPS tensor network. While for general TTNs not all tensors have a physical index, the following results can be straightforwardly generalized by just removing the physical index from the notation. Alternatively, every tensor without a physical index could be interpreted as having a dummy index with just a single entry corresponding to a single state, say $|0\rangle$, onto which the Hamiltonian acts as an identity $H|0\rangle = |0\rangle$. Note that if all sites have a physical index, the number of links is $L = N - 1$. In the following, we will often omit sums over auxiliary indices $\sum_{q_1 \cdots q_L}$ and assume Einstein convention for the summations.

An example for a TTN with $N = 7$ sites and $L = 6$ auxiliary indices (links) is shown in Fig. 1. The property distinguishing a TTN from a general tensor network is that the graph of a TTN is *loop-free*, i.e., to move from one site to any other there is only one unique path along the links. This also implies that by cutting any link, the tensor network splits into two disconnected segments. Therefore, at each link there is a notion of *left* and *right* which is a first hint towards the capability of TTNs to access the Schmidt decomposition and with it also the reduced density matrix as demonstrated below. We also define the *leaves* of the TTN as all tensors with just a single link index. For convenience, we assume site $N$ to be a leave of the TTN. Since TTNs are *loop-free*, one can also define a measure of distance $d_{ij}$ between two sites $i$ and $j$ given by the number of links one has to traverse to move from site $i$ to site $j$.

## 2.2 Tensor Gauge and Orthogonality Center

The representation of a quantum state as a tensor network is highly non-unique. This *gauge degree of freedom* can be used to obtain useful representations of the same quantum state as a TTN with certain properties, which can speed up calculations dramatically. As shown in Fig. 2, at each link one can insert an identity $\mathbb{1} = G \cdot G^{-1}$ for any invertible matrix $G$. By absorbing $G$ into one tensor and $G^{-1}$ into the other, a different representation of the same state is reached. In this part, we make use of this gauge degree of freedom to define an orthogonality center of the TTN.

A tensor $T_{Q_i}^{s_i}$ can be orthogonalized towards one of its neighbors with which it shares link $q_k$ as follows:

- Reshape $T_{Q_i}^{s_i}$ into a matrix $T_{(s_i, Q_i \setminus q_k),(q_k)}$ with rows $(s_i, Q_i \setminus q_k)$ and column $(q_k)$.

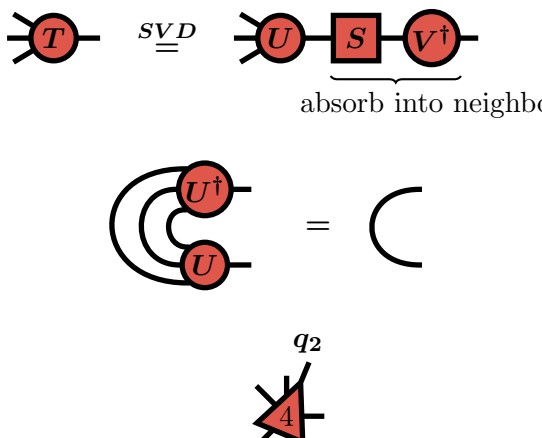

Figure 3: With an SVD, we can orthogonalize a tensor towards one of its neighbors with which it shares an index. *Top:* Tensor $T$ is reshaped into a matrix and the $U$-matrix of its SVD is used as new tensor. $S \cdot V^\dagger$ is absorbed into the neighboring tensor. *Middle:* Graphical representation of $U^\dagger \cdot U = \mathbb{1}$. *Bottom:* A tensor that is normalized towards one of its neighbors is depicted as a triangle pointing in the direction of this neighbor. The picture shows tensor 4 of the TTN in Fig. 1 orthogonalized towards tensor 2. Let us call the index connecting tensor 4 with tensor 2, $q_2$. In this case, we denote tensor 4 as $\left(T^{\mathcal{N}[q_2]}\right)^{s_4}_{Q_4}$.

- Perform an SVD (a QR decomposition is faster): $T_{(s_i,Q_i \backslash q_k),(q_k)} = \sum_\alpha U_{(s_i,Q_i \backslash q_k),(\alpha)} \cdot S_\alpha \cdot V^\dagger_{(\alpha),(q_k)}$.

- Keep $U_{(s_i,Q_i \backslash q_k),(\alpha)}$ as the new local tensor on site $i$ and absorb $S \cdot V^\dagger$ into the corresponding neighbor by multiplying $S \cdot V^\dagger$ onto it (formally also relabel $\alpha \to q_k$).

The SVD as well as the QR decomposition guarantees that the new site tensor has the property (see Fig. 3)

$$\left(U^\dagger \cdot U\right)_{(\alpha),(\alpha')} = \sum_{s_i,Q_i \backslash q_k} (U^\dagger)_{(\alpha),(s_i,Q_i \backslash q_k)} U_{(s_i,Q_i \backslash q_k),(\alpha')} = \delta_{(\alpha),(\alpha')}.$$

For tensors orthogonalized towards their neighbor along link $q_k$ we introduce the notation $\left(T^{\mathcal{N}[q_k]}\right)^{s_i}_{Q_i}$ (see Fig. 3 bottom).

As already mentioned, in TTNs there is a unique path between any two tensors. Therefore, by orthogonalizing a tensor towards one of its neighbors, we also orthogonalize it towards all other tensors, which can be reached via this neighbor. For example, to orthogonalize tensors 1, 2 and 6 in Fig. 1 towards tensor 3, we orthogonalize all of them towards tensor 4 with the procedure described above.

Next, let us introduce orthogonality centers. Site $i$ is an orthogonality center with tensor $C^{s_i}_{Q_i}$ if all tensors of all other sites are orthogonalized towards site $i$. To obtain such an orthogonality center, we can use the following algorithm:

1. Find the maximum distance $d_{max}$ between site $i$ and any other site in the TTN.

2. Initialize $d = d_{max}$ and perform the following steps until $d = 0$

   - Orthogonalize all sites $j$ that are at distance $d$ from site $i$ towards site $i$, i.e., towards the single neighbor on the path from $j$ to $i$.

   - Reduce $d$ by one $d \to d - 1$.

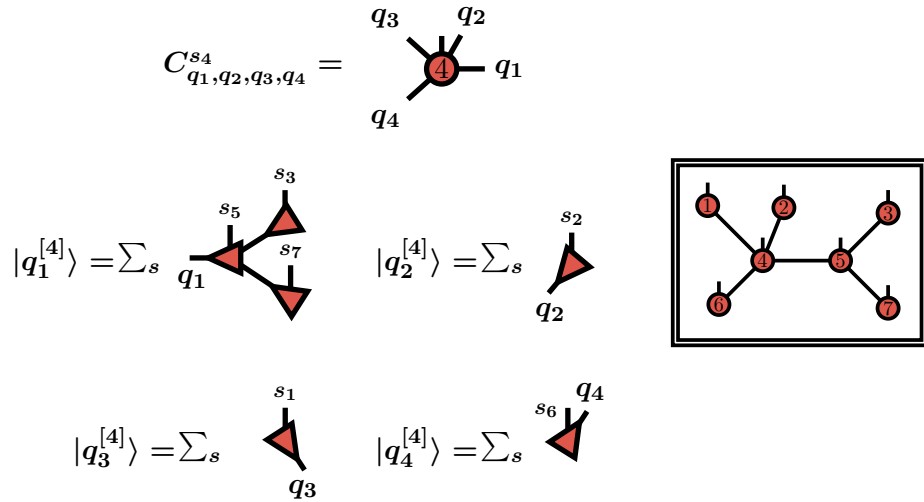

Figure 4: If the orthogonality center of the TTN depicted in Fig. 1 is placed on site 4, the center tensor $C$ has four link indices $q_1 \cdots q_4$. Each of these links corresponds to one of four mutually orthogonal set of states $\left|q_1^{[4]}\right\rangle \cdots \left|q_4^{[4]}\right\rangle$. This orthogonality is a direct result of the orthogonality property of the $U$-matrices of the SVD used on all sites except site 4 (see also Fig. 3). The inset is a reminder of the TTN used in this section.

For example, to orthogonalize the TTN shown in Fig. 1 towards site 4, we first orthogonalize sites 3 and 7 towards site 5 and then sites 1, 2, 6 and 5 towards site 4.

The wave function of a TTN with orthogonality center $C_{Q_i}^{s_i}$ can be written as:

$$|\psi\rangle = \sum_{\substack{q_1,q_2,\cdots,q_{r_i}\in Q_i \\ s_i}} C_{q_1,q_2,\cdots,q_{r_i}}^{s_i} |s_i\rangle \left|q_1^{[i]}\right\rangle \left|q_2^{[i]}\right\rangle \cdots \left|q_{r_i}^{[i]}\right\rangle$$

$$\left|q_k^{[i]}\right\rangle = \sum_{s_1\cdots s_r\in S_{q_k}^i} \left(T_{Q_1}^{s_1}\cdot T_{Q_2}^{s_2}\cdots T_{Q_r}^{s_r}\right)_{q_k} |s_1\cdots s_r\rangle$$

$$\langle q_k^{[i]}|q_k^{[i]'}\rangle = \delta_{q_k,q_k'}. \tag{3}$$

Here, $S_{q_k}^i$ is the segment of the tensor network that is obtained by cutting index $q_k$ and which does not contain site $i$. The states $\left|q_k^{[i]}\right\rangle$ form an orthogonal basis and are defined in Fig. 4 for the TTN of Fig. 1 with orthogonality center on site $i = 4$.

Orthogonality centers allow to easily calculate local observables acting on the orthogonality center. For example, the expectation value of the operator $\hat{A} = \sum_{s_i,s_i'} A^{s_i',s_i} \left|s_i'\right\rangle \langle s_i|$ acting nontrivially only on site $i$, reduces to:

$$\langle\psi|\hat{A}|\psi\rangle = \sum_{\substack{q_1 q_2\cdots q_{r_i}\in Q_i \\ s_i,s_i'}} \bar{C}_{q_1 q_2\cdots q_{r_i}}^{s_i'} \cdot A^{s_i' s_i} \cdot C_{q_1 q_2,\cdots q_{r_i}}^{s_i}, \tag{4}$$

where the bar denotes complex conjugation. Orthogonality centers hence reduce the costly contraction over the whole tensor network, to a simple contraction over the center tensor $C$ only.

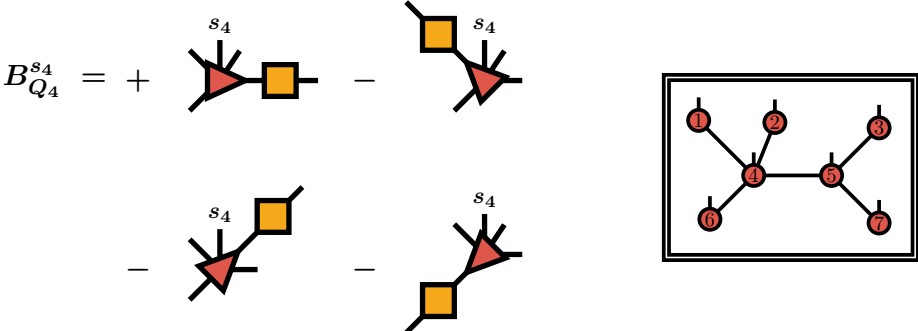

Figure 5: Definition of the vertical subspace for site 4 in Eq. 9 with end point site 7. The space of all tensors defining the kernel of the map from the tangent space to the physical hilbert space, is spanned by a matrix $X_{q'_l q_l}$ (yellow squares) for each link of the TTN. The physical index $s_4$ is labeled separately to distinguish it from the bond indices. The inset is a reminder of the TTN used in this section.

## 2.3 Truncation of TTNs

A TTN with an orthogonality center allows to calculate any Schmidt decomposition of the quantum state with respect to the two parts of the system defined by cutting any of the links of the orthogonality center. To do so, we reshape the center tensor $C$ into a matrix with physical index $s_i$ and one of the links $q_k$, combined into the row index and all other indices into the column indices, i.e., $C^{s_i}_{q_1, q_2, \cdots, q_{r_i}} = C_{(s_i, q_k),(Q_i \backslash q_k)}$. The Schmidt decomposition then follows from an SVD of this matrix:

$$C_{(s_i q_k),(Q_i \backslash q_k)} = \sum_\alpha U_{(s_i q_k),(\alpha)} \cdot S_\alpha \cdot (V)^\dagger_{(\alpha),(Q_i \backslash q_k)}$$

$$\Rightarrow |\psi\rangle = \sum_\alpha S_\alpha \underbrace{\sum_{s_i q_k} U_{(s_i q_k),(\alpha)} \left| q_k^{[i]} \right\rangle |s_i\rangle}_{|L\rangle_\alpha} \cdot \underbrace{\sum_{Q_i \backslash q_k} (V)^\dagger_{(\alpha),(Q_i \backslash q_k)} \bigotimes_{l \neq k} \left| q_l^{[i]} \right\rangle}_{|R\rangle_\alpha}$$

$$= \sum_\alpha S_\alpha |L\rangle_\alpha |R\rangle_\alpha. \tag{5}$$

Note that if one is interested solely in the truncation of a given orthogonality center and keeping it at the same site, an even more efficient approach would be to perform an SVD on the matrix $C_{(q_k),(s_i Q_i \backslash q_k)}$. Again, the orthogonality of the states $|q_k\rangle$ and the orthogonality of the $U$ and $V^\dagger$ matrices guarantee that the left and right vectors also form an orthogonal basis and hence Eq. 5 is a true Schmidt decomposition. Note that this Schmidt decomposition separates all sites in segment $S^i_{q_k}$ as well as site $i$ to the rest of the lattice. From there it is straightforward to calculate the reduced density matrix for one of these two subsystems and approximate states by keeping only the largest eigenvalues in the spirit of DMRG.

# 3 TDVP equations for Tree Tensor Networks

In this section, we generalize the derivation of the tangent space projector presented for MPS in Ref. [29] to general TTNs. While the overall approach is very similar to the derivation for MPS, the lack of a clear start and end point in the tensor network geometry will make the derivation and the subsequent integration of the equations quite different from standard MPS. TDVP amounts to the solution of the Schrödinger equation in the space spanned by the tensor

network without ever leaving this manifold (at least in its single-site variant). In TDVP, one solves a modified Schrödinger equation by projecting its right-hand side onto the so-called tangent space:

$$\frac{d \left|\psi[T]\right\rangle}{dt} = -i\mathcal{P}_{T_{\left|\psi[T]\right\rangle}} H \left|\psi[T]\right\rangle. \tag{6}$$

In the following, we want to find a representation of the tangent space projection operator $\mathcal{P}_{T_{\left|\psi[T]\right\rangle}}$, which not only depends on the current state $\left|\psi[T]\right\rangle$ but importantly also on the structure of the TTN.

### 3.1 Tangent Space Projector

Any element of the tangent space $\left|\Theta[B]\right\rangle$ is parametrized by a set of tensors $B_{Q_i}^{s_i}$:

$$\left|\Theta[B]\right\rangle = \sum_{i=1}^{N} B_{Q_i}^{s_i} \frac{d \left|\psi[T]\right\rangle}{dT_{Q_i}^{s_i}}. \tag{7}$$

Importantly, for each summand we use the representation of the state $\left|\psi[T]\right\rangle$ in which site $i$ is the orthogonality center, i.e., all tensors $T_{Q_j}^{s_j}$ are orthogonalized towards site $i$ such that

$$\left|\Theta[B]\right\rangle = \sum_{i=1}^{N} \sum_{\substack{Q_i \\ s_i}} B_{Q_i}^{s_i} \left|s_i\right\rangle \left|q_1^{[i]} \cdots q_{r_i}^{[i]}\right\rangle. \tag{8}$$

The gauge degree of freedom of the TTN reflects itself in the tangent space that not all linearly independent choices of $B_{Q_i}^{s_i}$ result in different tangent vectors. Ref. [29] solves this problem by first defining the so-called vertical subspace, i.e., all tensors $B_{Q_i}^{s_i}$ that give the zero-state $\left|\Theta[B]\right\rangle = 0$ and hence define the kernel of the map from the tensors to the physical Hilbert space. Then, imposing a gauge prescription, they fix this kernel to a single element which guarantees that the resulting parametrization is unique.

In order to arrive at a result that resembles the MPS algorithm, we first need to define a fixed end point of the TTN with the restriction that it should be a leave. Note however that any site of the tensor network can be used as end point. Without loss of generality, we choose site $N$ as end point. The vertical subspace, i.e., all tensors $B_{Q_i}^{s_i}$ for which $\left|\Theta[B]\right\rangle = 0$ can then be parametrized by matrices $X_{q'_k q_k}$ such that:

$$B_{Q_i}^{s_i} = \sum_{l=1}^{r_i} \sum_{q'_l} \left(T^{\mathcal{N}[q_l]}\right)_{q_1 \cdots q'_l \cdots q_{r_i}}^{s_i} X_{q'_l q_l} \cdot \mathrm{sgn}(q_l \to N)$$

$$\mathrm{sgn}(q_l \to N) = \begin{cases} 1, & \text{if } q_l \text{ points towards } N, \\ -1, & \text{otherwise.} \end{cases} \tag{9}$$

$\left(T^{\mathcal{N}[q_l]}\right)_{Q_i}^{s_i}$ is the unique tensor of the state $\left|\psi[T]\right\rangle$ with site $i$ orthogonalized towards the neighbor on the other end of the link $q_l$. This definition of the vertical subspace is depicted in Fig. 5 for the tensor $B_{Q_4}^{s_4}$.

The factor $\mathrm{sgn}(q_l \to N)$ is 1 if link $q_l$ points towards the end point and $-1$ otherwise. This construction guarantees that for any choice of $B_{Q_i}^{s_i}$ in the vertical subspace, $\left|\Theta[B]\right\rangle = 0$, because the single term with positive sign $(q_l \to N)$ is exactly canceled by one negative term of its neighbor (since there $(q_l \nrightarrow N)$). Note that this definition of the vertical subspace reduces in the case of MPS to the definition used in Ref. [29] if the right-most site of the MPS is chosen as the end point.

To uniquely specify the kernel, we impose the following matrix-valued (with indices $q_k$ and $q'_k$) gauge fixing condition for the $B$-tensors of the tangent space:

$$\sum_{\substack{Q_i \backslash q_k \\ s_i}} \bar{B}^{s_i}_{q_1 \cdots q'_k \cdots q_{r_i}} \cdot \left(T^{\mathcal{N}[q_k]}\right)^{s_i}_{q_1 \cdots q_k \cdots q_{r_i}} = 0, \qquad \forall \, i \neq N. \tag{10}$$

Again, the bar denotes complex conjugation. Above, $q_k$ is the single index pointing towards the end point $N$. These are $N-1$ matrix-valued constraints, for the $X$-matrices living on $L = N - 1$ indices. This implies that no ambiguity is left in the definition of the kernel, if we choose $B$-tensors according to Eq. 10.

It also guarantees that the overlap between two tangent vectors reduces to a contraction over local tensors only:

$$\langle \Theta[B'] | \Theta[B] \rangle = \sum_{i=1}^{N} \sum_{\substack{Q_i \\ s_i}} \bar{B}^{s_i}_{Q_i} \cdot B^{s_i}_{Q_i}. \tag{11}$$

Similar to MPS, we can now reformulate the projection problem of an arbitrary state $|\Xi\rangle$ onto the tangent space $|\Theta[B]\rangle = \mathcal{P}_{T_{|\psi[T]\rangle}} |\Xi\rangle$ as a minimization problem:

$$\min_{B} \big\| \, |\Theta[B]\rangle - |\Xi\rangle \, \big\|^2, \tag{12}$$

under the constraints given by Eq. 10. With Eq. 8 and using Lagrange multipliers $\lambda^{[i]}_{q_k q'_k}$ to account for the constraints, the minimization can be reformulated as:

$$\min_{B} \left[ \sum_{i=1}^{N} \sum_{\substack{Q_i \\ s_i}} \left( \bar{B}^{s_i}_{Q_i} \cdot B^{s_i}_{Q_i} - \bar{B}^{s_i}_{Q_i} \cdot F^{s_i}_{Q_i} - \bar{F}^{s_i}_{Q_i} \cdot B^{s_i}_{Q_i} \right) - \sum_{i=1}^{N-1} \sum_{q_k q'_k} \lambda^{[i]}_{q_k q'_k} \sum_{\substack{Q_i \backslash q_k \\ s_i}} \bar{B}^{s_i}_{q_1 \cdots q'_k \cdots q_{r_i}} \cdot \left(T^{\mathcal{N}[q_k]}\right)^{s_i}_{q_1 \cdots q_k \cdots q_{r_i}} \right], \tag{13}$$

with $F^{s_i}_{Q_i} = \langle s_i q_1 \cdots q_{r_i} | \Xi \rangle$. The solution to this minimization problem can be found by setting the derivatives with respect to $\bar{B}^{s_i}_{Q_i}$ as well as $\lambda^{[i]}_{q_k q'_k}$ to zero. Using some algebra we find the minimum for all sites $i \neq N$:

$$B^{s_i}_{Q_i} = F^{s_i}_{Q_i} - \sum_{\substack{Q''_i \backslash q''_k, q'_k \\ t}} \left(T^{\mathcal{N}[q_k]}\right)^{s_i}_{q_1 \cdots q'_k \cdots q_{r_i}} \left(T^{\mathcal{N}[q_k]}\right)^{t}_{q''_1 \cdots q'_k \cdots q''_{r_i}} \cdot F^{t}_{q''_1 \cdots q_k \cdots q''_{r_i}}, \tag{14}$$

while for $i = N$ it is just $B^{s_N}_{Q_N} = F^{s_N}_{Q_N}$. This allows us to obtain a representation of the tangent space projector $|\Theta[B]\rangle = \mathcal{P}_{T_{|\psi[T]\rangle}} |\Xi\rangle$ as:

$$\mathcal{P}_{T_{|\psi[T]\rangle}} = \sum_{i=1}^{N} \mathbb{1}_{s_i} \otimes \sum_{Q_i} \left| q_1^{[i]} \cdots q_{r_i}^{[i]} \right\rangle \left\langle q_1^{[i]} \cdots q_{r_i}^{[i]} \right| \quad - \sum_{<i,j>_{q_k}} \sum_{q_k q'_k} \left| q_k^{[j]\prime} \right\rangle \left\langle q_k^{[j]\prime} \right| \otimes \left| q_k^{[i]} \right\rangle \left\langle q_k^{[i]} \right|, \tag{15}$$

where $\sum_{<i,j>_{q_k}}$ denotes a sum over all nearest neighbors $i$ and $j$ with the corresponding index $q_k$ connecting these two sites. The graphical representation of the states in the second line of the tangent space projector for the bond connecting sites $i = 4$ and $j = 5$ is shown in Fig. 6. Formally, this result resembles the projection operator obtained for MPS [29]. The first term with positive sign corresponds to the forward time propagation of the site tensor. The second term on the other hand is the evolution backwards in time of the bonds between two site tensors and is a direct consequence of the gauge fixing of the tangent vectors used in Eq. 10.

$$|q_1^{[4]}\rangle = \sum_{s,q_1} \qquad |q_1^{[5]'}\rangle = \sum_s$$

Figure 6: Definition of the states used in the projection operator onto the link $q_k$ defined in the second line of Eq. 15 for link $q_1$ connecting sites $i = 4$ and $j = 5$.

## 3.2 Single-Site TDVP

With the representation of the projection operator in Eq. 15, we can go back to the projected time dependent Schrödinger equation (Eq. 6) and integrate each term one by one using Trotter breakups [39]. First, let us discuss a first-order update, which can later easily be modified to perform a second order integration. Since each term in the projection operator keeps all but one tensor fixed, the integration can be performed locally. Therefore, we define effective Hamiltonians for the sites $i$ and for the links $q_k$:

$$H_{(s_iQ_i),(s_i'Q_i')} = \left\langle s_i q_1^{[i]} \cdots q_{r_i}^{[i]} \middle| H \middle| s_i' q_1^{[i]'} \cdots q_{r_i}^{[i]'} \right\rangle \tag{16a}$$

$$K_{(q_k^{[i]}q_k^{[j]})(q_k^{[i]'}q_k^{[j]'})} = \left\langle q_k^{[i]} q_k^{[j]} \middle| H \middle| q_k^{[i]'} q_k^{[j]'} \right\rangle \tag{16b}$$

and solve equations of the form:

$$\dot{\mathbf{A}} = \pm i H^{\text{eff}} \cdot \mathbf{A}$$
$$\mathbf{A}(t + \Delta t) = e^{\pm i H^{\text{eff}} \Delta t} \mathbf{A}(t), \tag{17}$$

where $\mathbf{A}$ is either a site-tensor or a link tensor and $H^{\text{eff}}$ either $H_{(s_iQ_i),(s_i'Q_i')}$ (negative sign) or $K_{(q_k^{[i]}q_k^{[j]})(q_k^{[i]'}q_k^{[j]'})}$ (positive sign). In matrix form, the exponential of these effective Hamiltonians can be efficiently calculated using Krylov exponentiation.

A full TDVP step is then given by a series of $N - 1$ local updates of a site tensor and the corresponding link tensor connecting the site to the end point as shown below. The single local update on site $i$ and link $q_k$ is

- Orthogonalize the TTN such that site $i$ is the orthogonality center.

- Calculate the one-site effective Hamiltonian $H^{\text{eff}} = H_{(s_iQ_i),(s_i'Q_i')}$ (Eq. 16a) and forward time evolve (negative sign) according to Eq. 17 with $\mathbf{A} = C_{Q_i}^{s_i}$. If site $i$ is the chosen end point, stop here; otherwise continue.

- Reshape the time evolved tensor into a matrix $C_{Q_i}^{s_i} = C_{(s_iQ_i\backslash q_k),(q_k)}$ and perform an SVD (QR-decomposition suffices) $C_{(s_iQ_i\backslash q_k),(q_k)} = \sum_{q_k^{[i]}} U_{(s_iQ_i\backslash q_k),(q_k^{[i]})} \cdot \underbrace{S_{q_k^{[i]}} \cdot (V^\dagger)_{(q_k^{[i]}),(q_k)}}_{L_{q_k^{[i]}q_k}}$. As usual, take the $U$-tensor as new tensor on site $i$.

- Calculate the effective Hamiltonian $H^{\text{eff}} = K_{(q_k^{[i]}q_k^{[j]})(q_k^{[i]'}q_k^{[j]'})}$ (Eq. 16b) for link $q_k$. To do so, use the time evolved tensor obtained in the previous step for site $i$. Then evolve tensor $\mathbf{A} = L_{q_k^{[i]}q_k} \equiv L_{q_k^{[i]}q_k^{[j]}}$ from the previous step backwards in time (positive sign) according to Eq. 17. Finally, absorb the $C$-tensor onto the neighbor of site $i$ along $q_k$ by multiplying it onto its site tensor.

A full TDVP time step can then be achieved by the following sweeping procedure:

1. Choose a start and an end point; initialize site $i$ as the chosen start point.

2. Perform the following steps until $i$ is the chosen end point:

   - Find the link $q_k \in Q_i$ that connects site $i$ to the end point.
   - If any tensor attached to the other links $Q_i \setminus q_k$ has not been updated, choose one of these links and choose one of the leaves attached to the corresponding segment of the TTN as new site $i$.
   - Otherwise, perform a local update on site $i$ as described above and choose the neighbor of site $i$ along link $q_k$ as new site $i$.

3. Perform one last local update for the endpoint $i = N$ as described above.

A depiction of the sweeping order for the TTN in Fig. 1 is shown in Fig. 7. The procedure described above defines a first-order time step. A second-order method can easily be obtained by performing the first order time step with $\frac{\Delta t}{2}$ and then repeating the exact same steps in reverse order corresponding to repeated second order Trotter breakups $e^{\tau(A+B+C)} = e^{\frac{\tau}{2}C}e^{\frac{\tau}{2}B}e^{\tau A}e^{\frac{\tau}{2}B}e^{\frac{\tau}{2}C}$ used on Eq. 15. Importantly, this means that during the local update, the link update has to be performed *before* the site update (see also caption of Fig. 7).

Note that for a given TTN, there can be several versions of this algorithm depending on the sequence of chosen indices in step 2. Very often though, the TTN structure itself defines some natural order when to time evolve which sites, as we will see in the next section for the FTPS tensor network.

## 3.3 Two-Site TDVP

It is also straightforward to generalize the single-site TDVP approach presented above to a two-site TDVP integration scheme which allows to dynamically adapt the necessary bond dimensions. To do so, we need to define the two-site effective Hamiltonian $H^{\text{2-site}}$ for two sites $i$ and $j$ connected by the index $q_l$:

$$H^{\text{2-site}} = \left\langle s_i s_j Q_i^{\text{red}} Q_j^{\text{red}} \middle| H \middle| s_i' s_j' Q_i^{\text{red}} Q_j^{\text{red}} \right\rangle$$

$$\left| Q_i^{\text{red}} \right\rangle = \bigotimes_{q_n \in Q_i \setminus q_l} \left| q_n^{[i]} \right\rangle$$

$$\left| Q_j^{\text{red}} \right\rangle = \bigotimes_{q_n \in Q_j \setminus q_l} \left| q_n^{[j]} \right\rangle. \tag{18}$$

With this, only small modifications to the algorithm presented above are necessary. The single update for sites $i$ and $j$ sharing link $q_l$ becomes:

- Orthogonalize the TTN such that site $i$ is the orthogonality center.

- Calculate the two-site effective Hamiltonian $H^{\text{2-site}}$ according to Eq. 18 and forward time evolve (negative sign) with $\mathbf{A} = \sum_{q_l} C_{Q_i}^{s_i} T_{Q_j}^{s_j}$.

- Reshape the time evolved tensor into a matrix $A_{(s_i Q_i \setminus q_l),(s_j Q_j \setminus q_l)}$ and perform an SVD $A_{(s_i Q_i \setminus q_l),(s_j Q_j \setminus q_l)} = \sum_{q_l} U_{(s_i Q_i \setminus q_l),(q_l)} \cdot \underbrace{S_{q_l} \cdot (V^\dagger)_{(q_l),(s_j Q_j \setminus q_l)}}_{C_{Q_j}^{s_j}}$. In this step one can also trun-

  cate the smallest Schmidt values. As usual, keep the $U$-tensor to update site $i$ and $C_{Q_j}^{s_j}$ as site tensor on site $j$, shifting the orthogonality center to site $j$. If site $j$ is the chosen end point, stop here; otherwise continue.

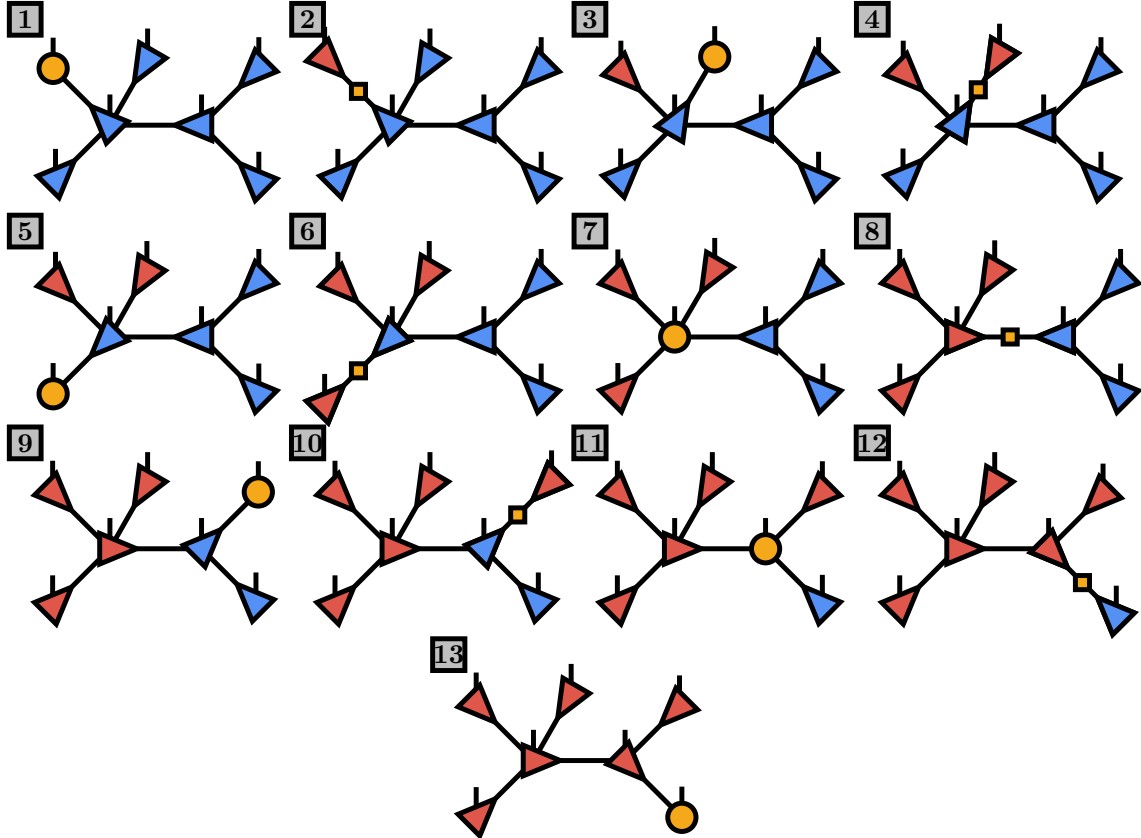

Figure 7: Update sequence to perform a first-order single-site TDVP time step from time $t$ to $t + \Delta t$ for the TTN shown in Fig. 1. Start point is site 1 and end point site 7. Yellow denotes tensor that are updated in the current step. Red and blue tensors indicate whether this tensor is taken at time $t + \Delta t$ (red) or $t$ (blue). Triangles indicate the orthogonalization of each tensor. Updates on site-tensors are in forward direction (negative sign), while updates on bond-tensors are backwards time evolutions (positive sign in Eq 17). For a second order update, first perform all steps $(1) \rightarrow (13)$ with time step $\frac{dt}{2}$ in the order shown and then reapply them in the reverse order $(13) \rightarrow (1)$, again with time step $\frac{dt}{2}$.

- Calculate the one-site effective Hamiltonian $H^{\text{eff}} = H_{(s_j Q_j),(s'_j Q'_j)}$ (Eq. 16a) for site $j$. To do so, use the time-evolved tensor obtained in the previous step for site $i$. Then evolve tensor $\mathbf{A} = C^{s_j}_{Q_j}$ backwards in time (positive sign in Eq. 17).

A full two-site TDVP step can then be performed by:

1. Choose a start and end point. Initialize site $i$ as the chosen start point.

2. Perform the following steps until $i$ is the chosen end point:

   - Find the link $q_k$ and the corresponding neighbor $j$ that connects site $i$ to the end point.

   - If any tensor attached to the other links $Q_i \setminus q_k$ has not been updated, choose one of these links and choose one of the leaves attached to the corresponding segment of the TTN as new site $i$.

   - Otherwise, perform a local update on site $i$ and $j$ as described above and go to site $j$, i.e., $i \rightarrow j$.

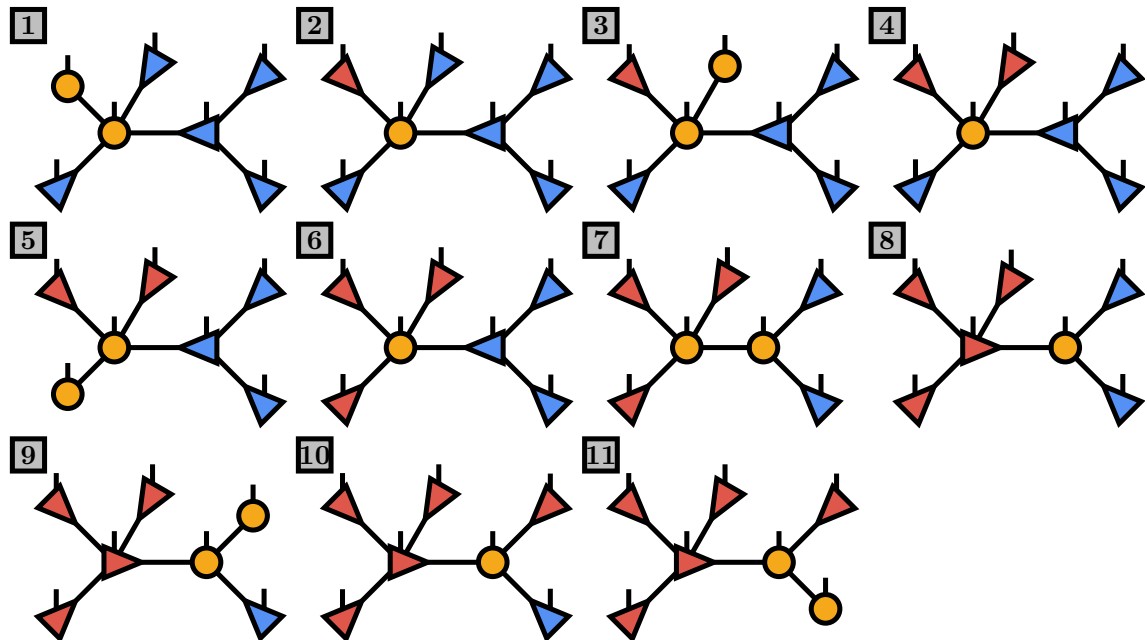

Figure 8: Update sequence to perform a first-order two-site TDVP time step from time $t$ to $t + \Delta t$ for the TTN shown in Fig. 1. Start point is site 1 and end point site 7. Yellow denotes tensors that are updated in the current step. Red and blue tensors indicate whether this tensor is taken at time $t + \Delta t$ (red) or $t$ (blue). Triangles indicate the orthogonalization of each tensor. Updates on two-site tensors are in forward direction (negative sign), while updates on a single site are backwards time evolutions (positive sign in Eq. 17). For a second-order update, first perform all steps $(1) \rightarrow (11)$ with time step $\frac{dt}{2}$ in the order shown and then reapply them in the reverse order $(11) \rightarrow (1)$, again with time step $\frac{dt}{2}$.

A depiction of the necessary sweeping order of this two-site scheme is shown in Fig. 8

## 4 TDVP for FTPS

An FTPS is a special TTN designed to efficiently encode states of multi-orbital Anderson Impurity Models (AIMs). An AIM consists of an interacting impurity coupled to a bath of free fermions with Hamiltonian

$$H = H_{\text{loc}} + H_{\text{bath}} + H_{\text{hyb}}$$
$$H_{\text{loc}} = \sum_{m\sigma} \epsilon_{m\sigma 0} n_{m\sigma 0} + H_{\text{int}}.$$
$$H_{\text{bath}} = \sum_{m\sigma} \sum_{k} \epsilon_{m\sigma k} n_{m\sigma k}$$
$$H_{\text{hyb}} = \sum_{m\sigma} \sum_{k} V_{m\sigma}^{[k]} \left( c_{m\sigma 0}^{\dagger} c_{m\sigma k} + \text{h.c.} \right). \tag{19}$$

$c_{m\sigma k}^{\dagger}$ ($c_{m\sigma k}$) creates (annihilates) an electron in chain $m$ with spin $\sigma$ on site $k$, where $k = 0$ denotes the impurity site (see Fig. 9 (b)). $n_{m\sigma k}$ are the corresponding particle number operators. $H_{\text{int}}$ is the interaction Hamiltonian that only couples impurity degrees of freedom and for which we choose the Kanamori Interaction [18,49] without the spin-flip and pair-hopping

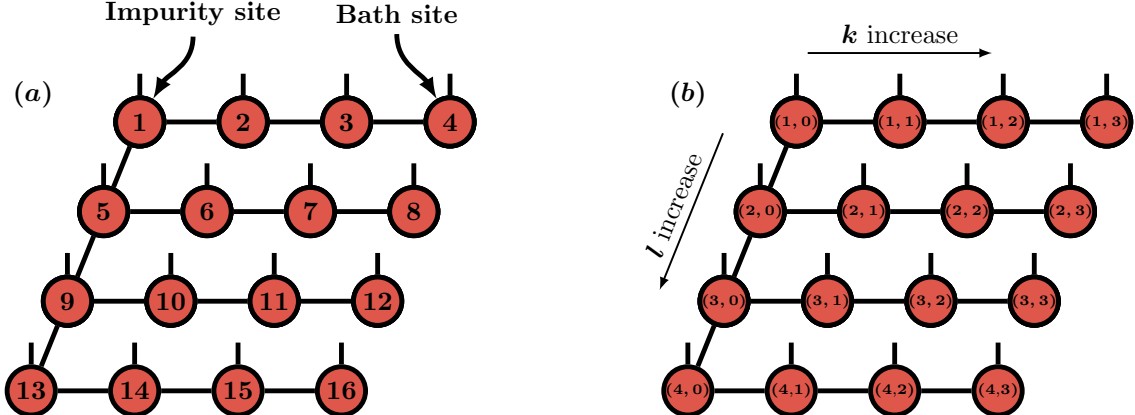

Figure 9: Graphical representation of a FTPS tensor network for a two orbital model. For each orbital, we use two chains, one for each spin-species. *(a)* one way to label the sites is just to numerate them in ascending order. *(b)* a different way to label sites is to specify the chain (orbital $m$ and spin $\sigma$) as well as an index (bath index $k$). This way to label sites resembles the labels used for the operators of the Hamiltonian in Eq. 19.

terms parametrized by two interaction strengths $U$ and $J$

$$H_{\text{int}} = U \sum_m n_{m\uparrow 0} n_{m\downarrow 0} + (U - 2J) \sum_{m' > m\sigma} n_{m\sigma 0} n_{m'\bar{\sigma}0} + (U - 3J) \sum_{m' > m\sigma} n_{m\sigma 0} n_{m'\sigma 0}, \quad (20)$$

where $\bar{\sigma}$ is the opposite spin direction of $\sigma$. In the following, we will use a combined index $l = (m\sigma)$ to denote the orbital and spin-degrees of freedom.

For a single orbital, an FTPS reduces to an MPS, while for multiple orbitals it has tensors with three link indices as depicted in Fig. 9 for a two-orbital model. It consists of a single MPS-like chain for the bath tensors of each orbital/spin and impurity tensors connecting the different chains. An FTPS for a $N_{\text{orb}}$-orbital AIM has a total of $N_C = 2N_{\text{orb}}$ chains. For simplicity we assume that each chain has the same number of bath sites $N_b$.

According to the algorithm presented in the previous section, we first need to choose a start and end point. We choose to start at the outermost bath site of the first chain (site 4 in Fig. 9 (a)) and the outermost bath site of the last chain as end point (site 16 in Fig. 9 (a)). To actually perform the time evolution, we choose to employ a hybrid TDVP scheme using 2-site TDVP for the bath tensors as well as for the bath-impurity link, and 1-site TDVP for the impurity tensors itself and the corresponding impurity-impurity links. We choose to use 1-site TDVP for the impurity links, since 2-site TDVP becomes computationally expensive, since one would have to deal with tensors with four link indices. This leads to the following algorithm for a single time step:

1. For $l = 1 : N_C - 1$ perform the following steps:

    - For $k = N_b : 1$:
        - Perform a two-site step on sites $i = (l, k)$ and $j = (l, k - 1)$ (see Fig. 9 for the definition of the site-labeling).
    - Perform a one-site step on the impurity tensor $i = (l, 0)$; $q_k$ connects site $(l + 1, 0)$

2. $l = N_C$, for $k = 0 : N_b - 1$ perform the following steps:

    - Perform a two-site step on sites $i = (l, k)$ and $j = (l, k + 1)$.

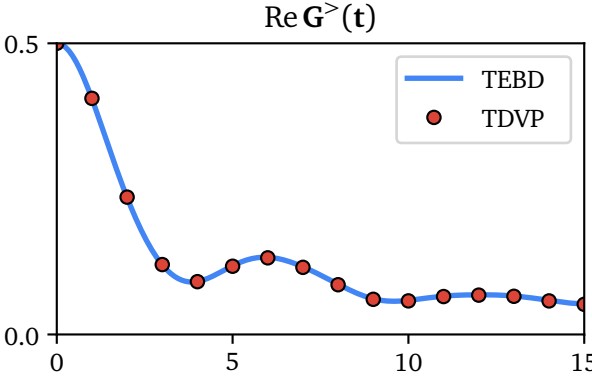

Figure 10: Comparison of the impurity greater Greens function $G^>(t)$ for a two orbital model between the TEBD time-evolution approach used in Ref. [18] and TDVP. The calculation was performed for a spin- and orbital degenerate model using 9 bath sites per orbital and spin with parameters $\epsilon_k = -0.8, -0.6, \cdots, 0.6, 0.8$ and $V_l^{[k]} = 0.1 \ \forall \ k, l$. Therefore, only diagonal entries of the Green's function are non-zero and for the diagonals there is only one independent function, i.e., $G^>_{l',l}(t) = \delta_{l'l} G^>(t)$. Interaction parameters were $U = 1$, $J = 0.1$. The impurity on-site energy was chosen to obtain particle hole symmetry, i.e., $\epsilon_{m\sigma0} = -\frac{3U-5J}{2}$. The time step for TEBD was $\Delta t = 0.01$ and for TDVP $\Delta t = 0.1$, since TDVP generally allows to use larger time steps [50]. Truncated weight (sum of all truncated Schmidt values) for TEBD was $10^{-12}$ without restricting the bond dimension, and for TDVP it was $10^{-9}$ for all links except the impurity-impurity links which were not truncated, but restricted to a maximal dimension of 50.

For the actual calculations, we apply the second order version of this algorithm by using only the half time step and reapplying each step in reverse order. Again, this also means that the order in the local updates changes. Note that the backwards propagation during the two-site update of the impurity site cancels with the subsequent forwards time evolution of the one-site step on the same impurity tensor. Therefore, these two steps can be omitted.

As a first demonstration of this algorithm, let us compare the TDVP time evolution to the TEBD-like approach used in Refs. [18, 19]. Therefore, we look at the greater Greens function of the impurity defined by:

$$G^>_{l',l}(t) = \langle\psi_0|c_{l'0}e^{-iHt}c_{l0}^\dagger|\psi_0\rangle e^{iE_0 t}. \tag{21}$$

$|\psi_0\rangle$ is the ground state of Hamiltonian $H$ with ground state energy $E_0$. For a degenerate two orbital model, Fig. 10 shows that the TDVP time evolution indeed produces the correct result. In a recent publication, the authors have shown, that for diagonal hybridizations, TDVP has larger errors than TEBD for the bath geometry chosen here [50]. This means that for such systems, the TEBD approach is most likely preferable over TDVP. For more involved baths on the other hand, TEBD can become difficult to formulate as discussed next.

One of the major advantages of TDVP is that it allows to perform the time evolution for arbitrary couplings in the Hamiltonian between the sites, as long as an *MPO* with the same tensor network structure as the state can be found. Eq. 19 is in fact not the most general AIM, since the bath only couples *diagonally* to its impurity. Often, one is also interested in so-called off-diagonal hybridizations which can be encoded as hoppings from impurity $l$ to a different bath $l'$. Therefore, we can account for off-diagonal hybridizations by replacing the hybridization

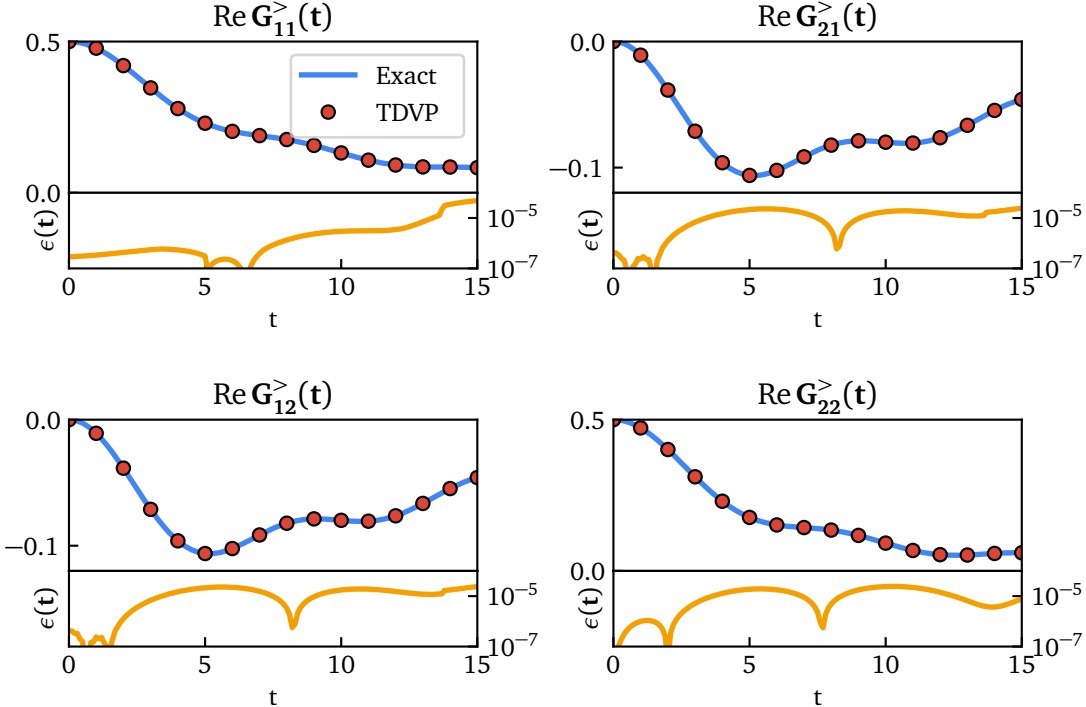

Figure 11: Comparison of impurity greater Greens function $G_{ll'}^{>}(t)$ for a two-orbital model between TDVP and the exact solution. In each segment, the top panel shows the Green's function itself, while the bottom panel shows the absolute value of the difference $\epsilon(t) = |G_{\text{exact}}^{>}(t) - G_{\text{TDVP}}^{>}(t)|$, i.e., the numerical error. We obtained the exact solution from diagonalization of the hopping matrix at $U = J = 0$. The calculation was performed for a spin-degenerate model using 9 bath sites per orbital and spin. We allowed off-diagonal hopping terms only between the orbital degrees of freedom, i.e., $V_{(m\sigma)(m'\sigma')}^{[k]} = \delta_{\sigma\sigma'} V_{mm'}^{[k]}$ and therefore $G_{(m\sigma)(m'\sigma')} = \delta_{\sigma\sigma'} G_{mm'}$. The parameters were $\epsilon_k = -0.8, -0.6, \cdots, 0.6, 0.8$ for all orbitals, diagonal hybridizations $V_{mm}^{[k]} = 0.1 \ \forall \ k, l$ and off-diagonal hybridizations $V_{mm'}^{[k]} = 0.05 \ \forall \ k$ for $m = 2, m' = 1$. These off-diagonal terms correspond to a hopping processes from the impurity of orbital 2 to the bath of orbital 1. The TDVP time step was chosen $\Delta t = 0.1$ and on-site energies were $\epsilon_{m\sigma 0} = 0$. Note that the off-diagonal hybridizations break the orbital degeneracy, albeit for the parameters chosen only slightly and the differences between the two orbitals are barely visible. Truncated weight (sum of all truncated Schmidt values) during DMRG and the time evolution was $10^{-9}$, except during the time evolution of the impurity impurity links where no truncation was performed. During DMRG as well as time evolution, the impurity-impurity links were restricted to 140.

terms in Eq. 19 with:

$$\sum_{ll'k} V_{ll'}^{[k]}(c_{l0}^{\dagger} c_{l'k} + \text{h.c.}). \tag{22}$$

It turns out that for each $k$, the matrix $V_{ll'}^{[k]}$ can be chosen as a lower-triangular matrix. This means, that for a spin-symmetric, two-orbital model there are three free parameters for each value of $k$ (instead of two for the diagonal hybridization).

As a second demonstration of the TDVP approach for FTPS we calculate the $2 \times 2$ matrix of the greater Green's function of such a spin symmetric two-orbital model. Since the TEBD

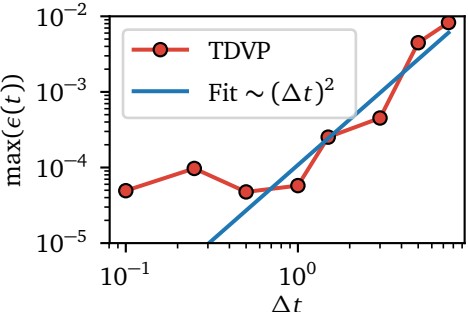 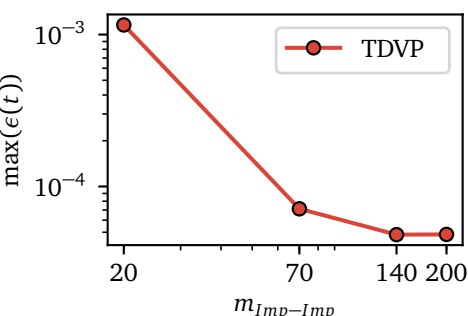

Figure 12: Error as a function of control parameters for the same AIM used in Fig. 11. We plot the maximum value of the error obtained in all four Green's functions $max(\epsilon(t))$.

*Left:* As function of step size $\Delta t$, the error shows the expected scaling $\sim (\Delta t)^2$ for larger values of $\Delta t$. The deviations for smaller values can be explained by the other sources of error, like the truncation during time evolution and a not perfect representation of the ground state. Additionally, we found TDVP in the star geometry to be quite sensitive to a too small time step in combination with a too large truncation. The parameters used in the truncation of the tensor network were exactly the same as discussed above.

*Right:* Error as a function of impurity-impurity bond dimension. All other parameters were the same as above.

approach we compared with in Fig. 10 is difficult to generalize to such off-diagonal hybridizations, we perform the calculation in the non-interacting case $U = J = 0$ and note that for tensor network based approaches this is a highly non-trivial situation. This is because the bipartitions defined by the links of the FTPS structure have non-trivial entanglement also for non-interacting systems, and the off-diagonal hoppings $V_{ll'}^{[k]}$ for $l \neq l'$ introduce entanglement *between* the orbitals, i.e., non-trivial links between the impurities. The results of such a comparison can be seen in Fig. 11. Having access to the exact solution, we also plot the difference between the exact and numerical Green's functions in the bottom panels. Again we find very good agreement between TDVP and the reference calculations.

Finally let us demonstrate that the results indeed converge with respect to the control parameters. The left plot of Fig. 12 shows the scaling of the error as a function of $\Delta t$ and we indeed observe the expected $\sim \Delta t^2$ behavior at larger values of $\Delta t$. The deviation of this behavior at smaller $\Delta t$ can be understood from the additional errors due to the truncations of the tensor network in the ground state as well as during time evolution. Additionally, we frequently observed that when TDVP is used in the star-geometry representation of the bath (with long-range couplings $V_{m\sigma}$), a good balance between truncation and $\Delta t$ is necessary. Surprisingly we found that it is often advantageous to use rather large time steps compared to what one would use in TEBD calculations. In the right plot of Fig. 12 we show the convergence of the error as a function of dimension of the impurity-impurity links which is usually the bottle-neck of FTPS calculations as these links need to transport the entanglement between the different orbitals $l$. Also here, we observe convergence with the control parameter, showing that TDVP indeed can be efficiently used to account for off-diagonal hybridizations.

# 5 Conclusion

We presented a generalization of the Time Dependent Variational Principle (TDVP) to general loop-free tensor networks (TTNs). The major advantage of TDVP over the commonly used TEBD approach is that the latter is often difficult to implement if long-range couplings are present in the Hamiltonian. TDVP on the other hand allows to perform the time evolution (either in imaginary- or real-time) for any Hamiltonian for which a representation in the same TTN structure can be found, which is often possible for long-range couplings. Using a similar derivation as in Ref. [29], we were able to find the projection operator onto the tangent space for any TTN - the central object in TDVP. Integrating the terms in the tangent space projector one after the other, equivalent to a Suzuki Trotter breakup, we were able to formulate TDVP in its single-site as well as two-site variant. We then applied TDVP to the FTPS tensor network which is a TTN especially suited for multi-orbital Anderson impurity models. For FTPS, TDVP is particularly appealing if there are off-diagonal hybridizations with the bath. In DMFT calculations, off-diagonal hybridizations are of significance to account for spin-orbit coupling effects as well as distortions of the crystal lattice. We verified the TDVP approach by comparing first to TEBD using a diagonal bath including interactions, and second to the exact solution in the non-interacting case for an off-diagonal bath.

When finalizing this manuscript we became aware of an independent publication by Kohn et al. [51], describing the TDVP applied to a TTN for periodic boundary conditions in a one-dimensional system.

## Acknowledgments

The authors would like to thank Florian Maislinger, Hans Gerd Evertz and Jutho Haegeman for fruitful discussions. This work was supported by the Austrian Science Fund (FWF) through the START program Y746, as well as by NAWI Graz.

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
