# Peer review of "Time Dependent Variational Principle for Tree Tensor Networks"

_SciPost Physics, doi:SciPost Phys. 8, 024 (2020)_

## Round 2 · Referee Report · Anonymous (Referee 1) · 2019-9-24

Strengths
Weaknesses
Report
The authors present a well-organized picture for running TDVP on TTN architectures. The technical part is clearly written, with in-depth details, such as the explanation of gauging and truncation. While the derivation of the tangent space projector is indeed technical, it follows mainly the original TDVP paper by Haegeman et al., so it should feel familiar to experts in the topic. The algorithms are clearly presented, which is a hands-on approach to programmers aiming to implement it, without the need to sift through all the maths. In the following we report a few remarks/ questions that the authors shuould consider:
-
In the introduction the authors state:
"One of the major reasons behind the success of tensor networks are the celebrated area lawsof entanglement [20] stating that the entanglement of ground states of gapped Hamiltoniansis proportional to the surface area connecting the two region" In the paper they cite (Ref.20) it states for the 1D case: "If a system is local and gapped, an area law always holds rigorously. Inmany specific models, prefactors can be computed. In contrast, if the interactions may be long ranged, area laws may be violated." Therefore, the authors may want to stress that such Hamiltonian also needs to be short range. -
The authors mention that TEBD is difficult to be used for off-diagonal hybridizations. In particular they state: "For more involved bathson the other hand, TEBD can become difficult to formulate as discussed next." However, it is not clearly explained why it is more difficult to use TEBD for off-diagonal hybridizations, since diagonal hybridization already requires long-range terms to be included. Could the authors briefly explain why off-diagonal hybridizations complicates the situation when using TEBD?
-
Could the authors provide the bond dimensions they used? We assume the authors converged their results with respect to the bond dimension. Did they see any differences in the convergence for sites where they use the one-site/two-site update? Since the one-site update does not allow for dynamical adjustment of the bond dimension, did the authors use an initial bond dimension larger than required for the ground state in order to have some overhead that might be needed by the time-evolution?
-
Since the authors implemented the second order integration, did they compare it to the first order method? Did they analyze the error as as function of the time step and find the expected scaling?
-
In addition to the references in the paper we deem the following papers to be relevant: SIAM J. Numer. Anal., 53(2), 917–941, (for TDVP in general) SIAM J. Matrix Anal. Appl., 34(2), 470–494 (for the derivation of the tangent space projector of a TTN)
-
When showing the single site TDVP the authors state: "Since each term in the projection operator keeps all but one tensor fixed, the integration can be performed locally." Could the authors explain briefly why the projection operator keeps all but one tensor fixed? After the Trotter breakup the projector and all tensors in the network are still time dependent, correct? If we understand correctly it is shown in SIAM J. Numer. Anal., 53(2), 917–941 (Section 4) and mentioned in the authors Ref.[29] that the integration can indeed be performed locally, but we think this is not obvious after the Trotter splitting of Eq.(6).
Finally, some minor remarks, such as typos we spotted:
-
The authors use both "loop-free" and "loop free", which should be homogenized.
-
FTPS abbreviation has not been introduced explicitely.
-
Fig.4 caption "Each of these links correspond" -> corresponds.
-
In section 2.2 the authors introduce notations for the orthogonality center (C) and for orthogonalized tensors. Maybe the authors want to consider using this notation also in the following to improve the presentation, for example: After Eq.(17), second $\bullet$, the tensor being evolved is an orthogonality center and could be called C instead of T.
We want to thank the Referee for the very positive Report. We incorporated the changes requested by the referee and answer to his/her questions below. Here are the answers to the Referee’s comments and questions:
** The Referee writes **
··· Could the authors briefly explain why off-diagonal hybridizations complicates the situation when using TEBD?
The problem with using TEBD for off-diagonal hybridizations is that it would require to use swap-gates (swapping sites in the tensor network) also for impurity sites. Then one would need to additionally swap the fermionic order of the tensor network involving an additional application of an MPO that acts non-trivially also on the bath sites ”between” the two impurities. While such an approach is possible, we found it to be rather inefficient compared to the clean formulation of TDVP.
** The Referee writes **
Could the authors provide the bond dimensions they used? We assume the authors converged their results with respect to the bond dimension.
For TEBD (Fig. 9, now Fig.10) we did not restrict the bond dimension and used a small truncated weight of 10−12. This resulted in a bond dimension of the ground state along the impurity-impurity links (the bottleneck of FTPS) of about 120. We performed TDVP for Fig. 9 using a truncation of 10−9 for fixed impurity-impurity bond dimensions m = 50 and m = 100, and show the result for m = 50 for which the result is converged on the scale shown in the figure (deviations < 10−3).
In Fig. 10, we used a truncated weight of 10−9 and restricted the bond dimen- sions of the impurity impurity links to 140, and checked the convergence of the results with a calculation using only 70. Note that the latter already provides very good results with errors about twice as large as the ones shown in Fig. 10. We also added this information to the plots in the manuscript.
** The Referee writes **
Did they see any differences in the convergence for sites where they use the one-site/two-site update?
We find that the error of the Green’s function on sites where we use the one-site update is a little larger than the error on sites where we use the two-site up- date. Such differences between physically equivalent sites are expected though, as already in the very FTPS tensor network structure treats sites that should be equivalent in-equivalently. For example in Fig. 9, the two equivalent impurity sites (1,0) and (2,0) are different in that (1,0) has two link indices while (2,0) has three.
** The Referee writes **
Since the one-site update does not allow for dynamical adjustment of the bond dimension, did the authors use an initial bond dimension larger than required for the ground state in order to have some overhead that might be needed by the time-evolution?
While this is generally the approach to take, here we decided to refrain from it. Instead we calculated the ground state already with very high precision to minimize the error from the DMRG calculation to be able to focus on the errors of the time evolution itself.
** The Referee writes **
Since the authors implemented the second order integration, did they compare it to the first order method? Did they analyze the error as as function of the time step and find the expected scaling?
We added a discussion and a plot demonstrating the expected scaling $∼ ∆t^2$ in the second order update. For the first order update we did not perform an in-depth comparison of the error as a function of time step.
** The Referee writes ** Could the authors explain briefly why the projection operator keeps all but one tensor fixed?
We answer this question in the pdf attached.
Attachment:
The authors have replied to my comments in detail, and I am satisfied with their answers.

Author: Daniel Bauernfeind on 2019-12-16 [id 683]
(in reply to Report 3 on 2019-10-20)We would like to thank the referee for his/her very positive and constructive feedback. We agree with the requested changes and changed our manuscript accordingly. Below we would like to answer a few specific points:
** The Referee writes: **
** Our Response: **
** The Referee writes: **
** Our Response: **
** The Referee writes: **
** Our Response: **
** The Referee writes: **
** Our Response: **

---

## Round 2 · Referee Report · Anonymous (Referee 2) · 2019-10-16

Strengths
1- well-presented work, introduction with the right context 2- technical details, every step can be followed by a reader who is familiar with tensor networks 3- benchmarking of the method is sound
Weaknesses
1- the discussion of the results is a bit rudimentary 2- the physical significance/interpretation of the results is not given
Report
A few remarks:
- The authors note that TEBD is hard for more complicated hamiltonians, especially long-range order; here, the works of Zaletel, et. al. should be mentioned as a possibility for extending the method to long-range interactions (Phys. Rev. B 91, 165112)
- One of major advantages (both numerical and conceptual) of TDVP over other methods is the fact that the energy (and, possibly, other preserved quantities) is preserved; is this the case for TTN as well? It would be good for the authors to comment on that, and check numerically that this is indeed the case (especially, when comparing with the TEBD results).
- Bond dimensions are nowhere mentioned in the numerical part; it would be good to have an idea of how large these have to be to attain reasonable timescales. Also, how do the different TDVP steps scale with the bond dimension?
- The authors say that "non-interacting systems are already entangled"; this is a confusing statement, since the amount of entanglement depends on what kind of partition one takes. It would be better to refer to real-space correlations that are non-trivial despite the system being non-interacting.
- Do the errors accumulate as time progresses? For spin chains, the bipartite entanglement entropy increases linearly as a function of time; is something similar happening here, and does the bond dimensions of the tensor network have to increase exponentially to keep a decent accuracy?
Requested changes
1- energy conservation and bond dimension as a function of time should be discussed (not necessarily new plots) 2- include references as discussed in the report
We would like to thank the second referee for the very positive feedback. We added the requested changes and answer the questions below.
The referee writes:
... that the energy (and, possibly, other preserved quantities) is preserved; is this the case for TTN as well?
Our response:
The conservation of these preserved quantities is only guaranteed in single-site TDVP and there is no reason to expect this to not hold for TTNs. For single-site TDVP, the conservation of energy is trivially fulfilled, since there is no truncation and the time-evolution is unitary. Other preserved quantities like the magnetization cannot change due to the abelian symmetry conservation used in the ITensor library. Therefore it is difficult to demonstrate the exact conservation of preserved quantities.
The referee writes:
The authors say that "non-interacting systems are already entangled"; this is a confusing statement, since the amount of entanglement depends on what kind of partition one takes ...
Our response:
The referee is correct, we clarified this statement, but we did not use the term "real-space" because it might be confusing. This is because we treat the impurity model in the so-called star geometry which is an energy representation and not a real-space one.
The referee writes:
Do the errors accumulate as time progresses? For spin chains, the bipartite entanglement entropy increases linearly as a function of time; is something similar happening here, and does the bond dimensions of the tensor network have to increase exponentially to keep a decent accuracy?
Our response:
In fact, we frequently observe that during the calculation of Green's functions of impurity problems, the bond dimension remains constant for quite some time after an initial increase. In other cases, it grows with time, but it rarely grows extremely fast. We think that this is because the application of annihilation/creation operators is closer to a local quench than to a global one. Such local quenches have been shown to often demonstrate entanglement dynamics $\sim \log{t}$.

---

## Round 2 · Referee Report · Anonymous (Referee 3) · 2019-10-20

Strengths
1- clearly written, pedagogic description of an algorithm which should prove useful for the community.
Weaknesses
1- the extension of the time dependent variational principle for matrix product states to tree tensor networks was already employed in the literature, though without an explicit description.
Report
The overall presentation is clear and well written. There are, however, several small details that are incorrect or imprecise and should be fixed. I provide the list of the issues I notice below, together with some comments which I hope would help to improve the presentation further.
Requested changes
1- The introduction gives a short list of time-evolution methods for tensor networks. For completeness, in the 1d case, authors should also mention Krylov-based methods (Zaletel, et al. Phys. Rev. B 91, 165112 (2015)), which, similarly as TDVP, can be used for long-range Hamiltonians, in particular on a tree. For PEPS, the current state-of-the-art for the infinite system is given by Czarnik et al. Phys. Rev. B 99, 035115 (2019) and Hubig et al., SciPost Phys. 6, 031 (2019). [if full update with recycling the environments cited as [33] is a minor update of TEBD than the algorithm presented in those two works is as well]
2- In Sec. 2.2, it is better (faster) to use QR decomposition in place of SVD. Also, the statement: "One of the major advantages of orthogonality center is that they ... speed up calculation of local observables" is hard to agree with. One has to orthogonalize the whole network to calculate single local observable in such a way -- which is quite costly (and re-orthogonalize to calculate the expectation value at another site)
3- In Sec. 2.3 in order to truncate leg q_k, one should use SVD on C_{(q_k), rest}
4- In Sec. 3.1, the authors write: "In order to arrive at a similar result, we first need to define a fixed starting and end-point with the restriction that these have to be a leave." Is this a necessary condition in this section? I don't see the first-site appearing in the derivation; nor the fact that end-point is a leaf. [not a leave, please double-check the spelling in the article]
5- In Sec. 3.2, the authors start by mentioning second-order Suzuki-Trotter decomposition. Then describe the 1-st order one in detail. I find it slightly confusing. The second-order decomposition is described later as a combination of first-one and its reversal. Additionally, having a second-order decomposition allows one to form 4th order or higher -- it might be worth mentioning this. Additionally, fix white spaces in Eq. 16
On page 12: The sentence "Importantly, this also means ... the order of a single and local update is reversed" -- is confusing. There are two single gates connected by a local link, one is going to be applied before, and the second after the local gate; It might not be obvious which single gate the authors refer to.
6- In Sec. 3.3, the authors write: "(QR-decomposition suffices)." Here one should only use SVD. Otherwise, the bond dimension of the link would grow exponentially with time. It would also be useful to explicitly mention that one performs truncation of smallest Schmidt values here.
7- In Sec. 4, please define H_{int}, so that the article is self-contained. In Fig 8, it would also be helpful, to add some arrows pointing in the direction in which l and k are changing; The statement 'connected to the impurity below' (bottom of page 14) only makes sense with respect to specific layout of that figure [I would suggest making it more figure-independent]
8- Combining 1-site TDVP and 2-site TDVP (bottom of page 14); Single site gates propagating the impurity sites backward (the second part of a 2-site step) and forward (first part of a 1-site step) are canceling each other. It is only necessary to evolve the links between impurity sites backward in this case -- it might be good to mention this.
9- I find the error discussion around Fig 10 highly lacking. The authors do not say, what is the bond dimension, which strategy do they apply to truncate it (in 2-site part of the scheme, is there some maximal bond dimension? or are the authors truncating the Schmidt values to some precision/discarded weights?) -- without such information the results cannot be reproduced. Also, since this is mostly a discussion of the method, I would appreciate the demonstration that this error (of 1e-5 in Fig 10) can be further controlled (i.e., everything is working as expected). What is the parameter of simulation which is ultimately controlling this error? (3-4 lines showing errors as this parameter is changed would be sufficient in my opinion). Also, the error would be more natural to read if the authors use a logarithmic scale in lower panels of Fig 10.
10- Figures; I like the idea of using the same small network for illustration of all the steps of derivation. However, having to refer to Fig 1 (a couple of pages back in some cases) is not optimal. I am wondering if some insets showing the layout of the whole network and site's numbering would help here?
11- Figure 7 is illustrating 1-site TDVP very nicely. I am wondering if a similar figure for the 2-site scheme (e.g., a second panel of fig 7) would not provide a good illustration of 3.3?

---

## Round 3 · List of Changes

We added "with short range couplings" in the introduction according to the first referee's remark.
We added the citations mentioned by all three referees. In particular, we mentioned the MPO-based time evolution by Zaletel et al. in the text and added a sentence about (SIAM J. Matrix Anal. Appl., 34(2), 470–494).
We added two sentences in the introduction mentioning Multi-layer Multi-Configurational Time-Dependent Hartree methods used in quantum chemistry and point out similarities to TDVP.
Changed SVD to QR decomposition in Sec. 2.2.
Changed the sentence regarding orthogonality centers allowing for easy calculation of local observables.
Rephrased the sentence about start- and end-points as requested by the third referee in Sec. 3.1.
Clarified the appearance of first- and second order Trotter decomposition. Clarified the sentence regarding the "reversed order of updates" in Sec. 3.2.
Removed "(QR decomposition suffices)" in Sec. 3.3 and mentioned that this is the point where the tensor network is truncated.
Added a figure showing the update sequence of two-site TDVP.
Added the example TTN to Fig. 4 and Fig. 5 to remind the reader of the structure.
Defined $H_{int}$ in Sec. 4.
Added a remark that a backwards time-step and a forwards step cancel in Sec. 4.
We added more information on the bond-dimensions and truncated weights to the captions of the figures in Sec. 4.
Improved the error discussion in Sec. 4, including a new figure, showing convergence with respect to the control parameters as well as the expected $\sim \Delta t^2$ scaling of the error in Sec. 4.
We clarified the sentence in Sec. 4 stating that "non-interacting systems are already entangled" as requested by the second referee.
Corrected typos.

---

## Editorial Decision

published